# Advancing Bilateral Limbal Deficiency Surgery: A Comprehensive Review of Innovations with Mucosal Cells

**DOI:** 10.3390/biomedicines13030630

**Published:** 2025-03-05

**Authors:** Zahra Bibak-Bejandi, Mohammad Soleimani, Zohreh Arabpour, Emine Esra Karaca, Elmira Jalilian, Hassan Asadigandomani, Reyhaneh Bibak-Bejandi, Ali R. D’jalilian

**Affiliations:** 1Department of Ophthalmology and Visual Sciences, University of Illinois at Chicago, Chicago, IL 60612, USA; zhrbibak@uic.edu (Z.B.-B.); soleimani_md@yahoo.com (M.S.); arabpour@uic.edu (Z.A.); dremineesra@gmail.com (E.E.K.); jalilian@uic.edu (E.J.); 2Department of Ophthalmology, Ankara Bilkent City Hospital, University of Health Sciences, Ankara 06800, Turkey; 3Farabi Eye Hospital, Tehran 1336616351, Iran; hasanasadigandomani0800@gmail.com; 4Hazrat Fatemeh Hospital, Tehran 4818813371, Iran; reyhanebejandi@gmail.com

**Keywords:** bilateral limbal stem cell deficiency (BLSCD), decellularized limbus, cultivated oral mucosal transplantation (COMT), cell therapy, simple oral mucosal transplantation (SOMT), mucosal cell, oral mucosal graft transplantation (OMGT)

## Abstract

Besides alternative surgical methods for bilateral limbal deficiency, such as KLAL (keratolimbal allograft), living-related conjunctival limbal allograft (LR-CLAL), and keratoprosthesis, regenerative medicine often necessitates the use of alternative sources of limbal cells in cases where access to fellow eye source cells is limited. Mucosal cells are most commonly used to restore limbal tissue in such scenarios. Current techniques involving mucosal cells include cultivated oral mucosal transplantation (COMT), oral mucosal graft transplantation (OMGT), and simple oral mucosal transplantation (SOMT). COMT requires suspension of cells and a culturing process that is time-consuming and cost-prohibitive. In contrast, OMGT requires solely a strip of mucosal graft for transplanting into the deficient eye. The most recently developed practice, SOMT, in which chopped biopsy tissue is transplanted into the deficient area, compensates for problems associated with both COMT and OMGT, making the process of addressing bilateral limbal deficiency easy, time-saving, and affordable. Although some undesirable outcomes, such as angiogenesis, can occur post-transplantation, and the ultimate goal of differentiation into limbal epithelial stem cells may not be achieved, mucosal cell sources can be a good alternative for stabilizing the ocular surface. Some studies emphasize that co-culturing limbal niches in mucosal cell cultures can enhance differentiation capability. This concept highlights the importance of the limbal environment in the differentiation process. In this review, we demonstrate the ongoing changes in surgical technique trends and how they have made mucosal cell transplantation easier and more effective for limbal regeneration.

## 1. Introduction

The limbus is a ring-shaped zone that hosts the corneal epithelial stem cells. Corneal epithelial stem cells play a key role in the maintenance of a healthy and clear cornea, which remains essential for clear vision [1]. Any event in this critical area, whether caused by mechanical, chemical, or thermal trauma or by an immunological disorder, can destroy these cells and result in a condition known as limbal deficiency [2]. Conjunctivalization and neovascularization of the cornea can disrupt its transparency, possibly resulting in near-total blindness. The surgical technique selected to address limbal deficiency varies depending on unilateral or bilateral eye involvement. In cases where only one eye is affected, strategies typically rely on limbal cell sources from the healthy eye, including such as conjunctival–limbal autograft (CLAU), cultivated limbal epithelial transplantation (CLET), and the latest development of simple limbal epithelial transplant (SLET) [3,4,5]. CLAU and CLET, while having high success rates, come with unique risks and pitfalls. Due to the biopsy size involved in CLAU, limbal deficiency may result in the previously healthy eye [6], and, while CLET utilizes a significantly smaller biopsy size, its process, which depends on time-consuming culturing, limits its utility. Within the past 15 years, to circumvent these obstacles, SLET was introduced in 2012, decreasing the lengthy time restrictions and costs of CLET and improving the simplicity of transplantation [7].

In a bilateral limbal deficiency situation, deprivation of any autologous limbal cell source requires the scientists to rely on other surgical techniques. In such cases, keratolimbal allograft (KLAL) is a common treatment modality, involving the transplantation of the superficial limbus from a donor eye onto the affected area. However, the reliance on immune suppression and the risk of graft rejection has made this approach less favorable [8]. In addition, different types of extra limbal sources with the ability to differentiate into limbal epithelial stem cells have been used, such as epidermal, hair follicle, and mesenchymal stem cells, as well as the most commonly used source, a mucosal cell [9,10]. Cultivated oral mucosal epithelial sheet transplantation showed cell markers comparable to limbal epithelial cultured cells, emphasizing the capability and potential of these cell sources. In this process, small mucosal biopsies were harvested and treated with dispase, with the resulting cell suspension then co-cultured with feeder cells [11], with or without base membrane [12]. Despite the positive results in clinical use [13], the culturing processes are time-consuming and generally less cost-effective. To compensate for these barriers, mucosal grafts and simple oral mucosal cell transplantation have been recently used [14,15].

In oral mucosal graft transplantation, the strip of the mucosal graft is transplanted onto the limbal deficient area, requiring less complicated preparation and improving costs [15]. For example, the crescent labial mucosal was trephined to determine the depth and border. A 5 mm (width) × 250 μm (depth) mucosal graft was then detached using a Beaver mini-blade and transferred to the peripheral cornea. First, the inner side was sutured with nylon, followed by stabilization of the graft on the limbus using tissue glue. The outer edge was anchored with sutures to the outer side [16]. Using the vacuum trephine helped determine the depth and create a more desirable graft that was thin and free of attached tissue [16]. In a simple oral mucosal epithelial transplant (SOMET), the chopped tissue is transplanted on the deficient area, which is fixed in different ways such as with fibrin glue and contact lens [14,17,18]. In this method, after extracting the mucosa and submucosal epithelial layer, the tissue is soaked in dispase to help remove the subepithelial residue. The chopped pieces are then transplanted and fixed in place with a contact lens [19]. Some studies have shown that co-culturing limbal niche cells (LNCs) with mucosal cell cultures reduces angiogenesis and complications, highlighting the crucial role of the limbal niche environment in promoting healing [20]. Although techniques for utilizing mucosal cell sources are more varied and have simplified procedures, particularly in SOMET, scaffolds, and the limbal niche, remain critical in limbal reconstruction [21]. They support differentiation and provide innate homeostasis in patients with limbal deficiency [21]. In this context, we introduce acellular limbus scaffolds as a novel three-dimensional scaffold for limbal reconstruction [22]. Combining simplified methods like SLET and SOMET could offer innovative solutions for limbal reconstruction. Hence, this review aims to assess the evolving trends in surgical techniques that utilize mucosal cells and new acellularized limbal scaffolds.

## 2. Cultivated Oral Mucosal Epithelial Sheets

While CLET is effective in cases of unilateral limbal deficiency, for bilateral limbal deficiency, oral mucosal cells are often the preferred source due to their ease of access and successful outcomes [23]. Comparison of the cultured limbal and mucosal cells indicate their similar morphology and putative stem cell markers, expressing the markers Keratin 3 (K3), delta Np63, membrane-associated mucins (MUC 1, 4, and 16), ABCG2, and p63, as well as p63 isoforms [24]. Although the clinical outcomes of limbal cell applications are more acceptable and the success rate of allogeneic cultured limbal epithelial transplantation (ACLET) remains higher at 71.4% compared with 52.9% for cultivated oral mucosal epithelial transplantation (COMET) [25], mucosal cells can be an appropriate alternative instead of KLAL in cases of bilateral limbal deficiency, which requires long-term immunosuppression [26,27]. For instance, p63-positive cells in the epithelial layer were exhibited for at least eight months after the transplantation in rat models [28]. Apart from favorable clinical outcomes, epithelial stem cell marker p63 has been detected in all patients who have undergone COMET and is another positive sign of a successful treatment. Unique corneal epithelial markers were predominantly expressed in three patients and weakly in two patients with epithelial tight junction protein 1 (ZO-1) [11]. In addition to the mentioned biomarkers, a well-organized epithelium layer was represented in 76.4% of 17 aniridic patients who underwent COMET [29]. Application of this method in a limbal deficient model led to the desired clinical outcome with low levels of MMP-3, a pro-angiogenic factor, and higher TIMP-3 levels, an anti-angiogenic factor [30]. In a different study, corneal homeostasis was also established [31], and similarly, Prabhasawat et al. outlined rates of 79.3% and 70.5% of corneal reconstruction within a 1- and 7-year post-transplantation follow-up [32], which were more promising than the 64.8% and 53.1% within 1 and 3 years found by Satake et al. [33].

Preservation of the cultured autologous oral mucosa epithelial cell sheets (CAOMECs) in the liquid nitrogen (LN2) over a long period, with the preservation of phenotype and differentiation markers makes it more valuable for transporting tissues into far regions [34]. Similar colony-forming efficiency (CFE), cell proliferation, and stem cell markers in both Stevens–Johnson syndrome (SJS) and non-SJS mucosal sources for COMECs indicate its potential as a source in autoimmune diseases [35]. Gene-edited mucosal cell transplantation via siRNA mediation represents an alternative treatment in ectodermal dysplasia patients [36], and it could serve as a key for autologous cell therapy in patients suffering from genetic malformations. All these aspects represent the multipotential applicability of mucosal cells in various limbal deficiency patients.

After pannus, conjunctiva, and symblepharon resection, the COMECs offer another option for transplantation to a denuded limbal deficient area of the stroma. To prepare sheets, after mucosal excision, the subepithelial region can be separated by the dispase, after which epithelial cells can be separated by trypsin and ethylenediaminetetraacetic acid (EDTA). The suspension of isolated epithelial cells can be co-cultured with the feeder cells [11], but it may take 2–4 weeks to reach COMECs [37].

There are two approaches for culturing: carrier-free (including enzymatic and thermo-responsive) and carrier-dependent. The cultured cell sheet may be detached from the thermo-responsive culture surface by decreasing the temperature to below 20 °C for 30 min [12]. The harvested cell sheet immunofluorescence evaluation was positive for putative stem cells p63, ΔNp63, and β1-Integrin at the basal layer. Transplantation on the deficient model led to stratified and transparent layers on the cornea [38]. Instead of employing a temperature-responsive polymer for detachment, collagenase could be used as an alternative to form an adhesive sheet comprising stratified and living cells on the deficiency model [39]. It could also be used as a clinical grade detachment factor [39]. Comparable features of mucosal cell sheets, such as proliferation, differentiation, and preservation of putative stem cell markers, could make this approach more valuable when comparing dispase and fibrin support as detachment occurs [39].

For the first time, COMECs on the amniotic membrane (AM) were presented by Nakamura and Kinoshita, which were subsequently transplanted onto the rabbit limbal deficiency model, resulting in stratified clear vision [37]. Transmission electron microscopy (TEM) and immunocytochemistry after mucosal cells cultured on the AM confirmed the stratified layers markers, including connexin 43, K3, K4, and K13, as well as progenitors’ markers at basal cells, including β1-integrin/CD29, ABCG2, p63, and p75 [40]. While AM remains the predominant carrier for cell culturing in limbal stem cell deficiency (LSCD) disease, numerous studies suggest biosynthetic scaffolds as a viable alternative carrier. Hyaluronan (HA) hydrogel scaffolds are one of these alternative carriers that consist of hyaluronic acid and play a vital role in the formation of the extracellular matrix (ECM). HA is also applied for culturing the human corneal endothelial cells (hCECs) [41] and human lens epithelial cells (hLECs) [42]. In another study, researchers coated the HA hydrogel scaffold with collagen IV, one of the essential proteins in ECM, to improve cell attachment and create full COMECs. The ∆Np63α and ABCG2 gene expression increased relative to control, indicating success in this certain procedure. Additionally, there was an increase in the integrin-αV expression, which is associated with scarless wound healing, and cadherin-1, which protects the ocular surface [43].

Another alternative was real architecture for 3D tissue (RAFT) based on type I collagen, which was developed by Anna R. O’Callaghan and provided compatibility with oral mucosal cells and limbal epithelial cells [44]. Lotrafilcon, a contact lens, provides a scaffold for culturing mucosal cells. FDA-approved and easily operated, its putative stem cell marker expression (vimentin, p63, and CK14) at the basal layer makes it applicable for transplantation [41]. Fibrin, which aligns with Good Manufacturing Practice (GMP) and operates independently of xenogeneic culturing materials, serves as a scaffold for mucosal cell culturing. The addition of tranexamic acid (TA) limits its early degradation. Furthermore, the expression of p63a and CK3 markers at the basal and superficial layer confirms the maintenance of mucosal cell capacity [42]. As a common method, animal-derived growth supplements applied for co-culturing material in CAOMECS may lead to complications. To prevent immunologic rejection and infection transmission, the KaFa™ medium, formulated for clinical-grade corneal epithelial cell sheets, was applied without any feeder cell and animal-oriented medium (AOM). The cells were grown on CELLstart™, the GMP-grade ECM. Epithelial progenitor cells (deltaNp63), corneal marker K3/K12, and E-cadherin expression were similar in a group of cG-CAOMECS and CAOMECS (AOM + feeder cells) [44]. For providing GMP conditions for culturing, the human amniotic membrane (HAM), human autologous serum (HAS), and fetal calf serum (FCS) were replaced without any animal products, which represent DNp63a, ABCG2, and C/EBPd as putative stem cell markers. The clinical usage of this sheet in two patients represents the integrity and homogeneity of the ocular surface [43].

Human dermal fibroblasts (DF) and mouse 3T3 fibroblasts, used as feeder cells, were co-cultured with both human limbal and oral epithelium cultures. This resulted in higher expression of Iβ1 and p63 markers, as well as a higher colony-forming efficiency (CFE) level, compared to human mesenchymal stem cells (MSC) [45]. In the culture of both mucosal and limbal cells, human oral mucosal fibroblasts (hOMF) can act as feeder cells, creating a stratified layer with a stem cell phenotype at the basal layer and squamous cells at the upper layer [44]. Besides the scaffold, feeder cell, and culturing condition, different locations of mucosa and culture media can affect the differentiation capacity, lower lip, and transition zone, indicating more capacity growth [46]. Transplanted buccal epithelial stem cells (BESCs) into AM showed a high nuclear/cytoplasmic (N/C) ratio and positive melanoma-associated chondroitin sulfate proteoglycan and connexin-43, allowing a visual acuity (VA) of 3/10 [47]. Stem cells from rabbit labial mucosa (rLMSCs), placed onto a denuded amniotic membrane (d-AM), were experimented with in two different ways on a rabbit model with limbal deficiency: either with the epithelium side facing downwards or upwards, compared to d-AM alone as a control group. Ninety days post-transplantation, the arrangement with the cell side facing up and the membrane side down exhibited the greatest corneal epithelialization and reduced opacity and neovascularization [48].

In a variety of limbal deficiency scenarios, such as SJS, thermal and chemical injuries, and ocular cicatricial pemphigoid (OCP), COMET has been utilized. Keratoplasty could be applied in some cases due to stromal infiltration [49]. Best-corrected visual acuity (BCVA) increased from 2.67 ± 0.08 to 0.64 ± 0.27 Log MAR among 14 patients following penetrating keratoplasty (PKP) after COMET [50]. On the other hand, in a recent study, corneal button immunofluorescence evaluation after COMET was positive for PAX6, K12 expression (cornea-specific markers) at the basal layer, and proliferation marker (Ki-67) at the conjunctivalization layer, confirming the transmission, stability, and differentiation ability of mucosal cells [51]. CK3 and CK12 positive markers revealed the presence of mucosal cells and corneal cells in successful patients. Total conjunctivalization occurred in the failure group and expressed conjunctival (CK7) and oral mucosal epithelial (CK3) markers after long-term following [52]. Among all the mentioned markers, SOX2 is represented as a specific marker to differentiate from other ocular surface cells [53]. In another study, decreases in conjunctivalization and symblepharon, as well as improvements in BCVA in 95% of patients, were seen in long-term monitoring post-COMET [13]. Improvement in the total ocular surface grading score from 7.0 to 3.0 during 6 months of follow-up after severe inflammation demonstrated the long-term stability of the treatment [54].

To reach a more feasible and less equipped method for limbal regeneration, transdifferentiated oral mucosal epithelial cells (T-OMECs), at 12 and 6 o’clock positions, about 2 mm posterior of the limbus, were injected sub-conjunctively in the limbal deficient model with epithelial tracking by the red fluorescence detected at the limbus in day 14 and at corneal at day 28. Desirable clinical outcomes and maintaining the corneal-like phenotype highlighted the route’s applicability [55]. The T-OMECs express Pax6 and CK12 in the central cornea and limbal epithelium, while the oral mucosal epithelium predominantly expresses CK13 instead. As a result of co-culturing with limbal niche cells, T-OMECs show an increased expression of Pax6 and CK12 and a reduced expression of CK13. As a result of co-culturing with limbal niche cells (LNCs), T-OMECs show an increased expression of Pax6 and CK12 and a reduced expression of CK13. Although these promising results were demonstrated, stability and cell survival require further long-term studies [55].

## 3. A Direct Autologous Oral Mucosal Epithelium Graft Transplantation

For the treatment of trachomatous pannus, and acute and chronic ocular burns in single-eye patients, Deing has introduced direct autologous oral mucosal epithelium graft transplantation [56]. Harvesting and transplantation are the two main steps in this technique. In direct autologous oral mucosal epithelium graft transplantation, half to two-thirds of the mucosal graft is placed at the corneal site and the rest of that suture at the corneo-limbal region after tissue excision [57]. Harvesting and preparation of the mucosal graft is a pivotal step to achieve the well-adjusted mucosal–corneal attachment without stepping, which is modified in many ways using the dispase II to facilitate the separation of the substantia propria initiated by Gipson to increase the healing process [57]. Recent studies also insist on this step for epithelial separation [19]. To have determined depth and circumferential graft, replacing the vacuum trephination proves its applicability for matching the limbal curve [16]. After 4 months, circumferentially trephined autologous oral mucosal graft transplantation improved VA to 0.526 ± 0.470. In four patients, results showed a decrease in corneal surface erosion and neovascularization, as well as a decrease in stromal opacity in two patients [16]. A woman with bilateral limbal deficiency from SJS, who initially had only light perception in her VA, underwent symblepharon release and mucosal membrane graft (MMG) in her left eye. The thinly sliced mucosal graft was stitched in a circular pattern 1 mm behind the limbus. One month later, her VA improved to 20/160 with corneal clarity. Five months later, keratinization was removed, and lid margin MMG was performed. The use of a scleral lens further enhanced her vision [58]. In seven cases, oral mucosal transplantation reduced pain and photophobia and improved VA for all of the patients [15]. After oral mucosal epithelial transplantation (OMCT) in 49 eyes suffering from various limbal deficiencies, including SJS, explosive injuries, thermal and chemical burns, and multiple pterygiums, results showed 53.06% of patients with cornea transparency, 95.92% with improvement in BCVA, 89.80% with stable epithelium until the final follow-up, and 89.80% experiencing a reduction in neovascularization [59]. Prepared labial mucosal was sutured around the limbus on the established ex vivo corneal model. After 21 days, the mucosal graft integrated with the membrane, with cell outgrowth over the cornea and positive mucosal epithelial keratin markers, demonstrating the effectiveness of the mucosal graft in limbal regeneration [60].

## 4. Simple Limbal Epithelial Cells Transplantation (SLET) and Simple Oral Mucosal Epithelial Transplantation (SOMET)

A new approach is proposed to address the disadvantages of conventional methods, such as induction of limbal deficiency, resulting from biopsy size in CLAU, and introduce a more cost-effective and time-saving method than CLET [61]. SLET has been introduced as a new method for the transplantation of stem cells by Sangwan et al. A tiny biopsy of the limbus was excised and chopped into six to eight pieces and then glued on the AM on the limbal deficient eye [7]. Due to the risk of limbal biopsy loss in 2020, glue-less simple limbal epithelial transplantation (G-SLET), developed by Boris E. Malyugin, tiny pieces of the healthy limbus were implanted in radial symmetrical superficial incisions on the targeted eye with AM suturing around the limbus. Post-surgery follow-up confirmed the static location of the tissue and semitransparent epithelium covering the cornea [17]. The mucosal graft can be harvested manually or using trypsinization and scissors [62]. Inspired by the SLET, simple oral mucosal epithelial transplantation (SOMET) was applied for a chemical burn, a lower lip mucosal graft was harvested, and a submucosal graft was trimmed manually, and then small chopped segments were circularly glued on the AM, second, the AM was covered with the tissue segment using fibrin glue, and then secured by contact lenses. BCVA reached 1/10 at the month 13 follow-up [14]. In another case, a premature male infant with bilateral peribulbar dermoid underwent dermoid resection. After a few months candidates for SOMET, 3/4 portion mucus membrane suture on the AM, with 1-year follow-up revealing visual behavior improvement [62].

During the process, mucosal sheets are digested in dispase for 60 min and transferred to the denuded rabbit cornea as a scaffold, which had previously been placed on the cylindrical agarose gel for 2 weeks. A stemness-positive layer attachment to the stroma (δNp63 positive cells) insists on the proliferation and differentiation potentiality of the mucosal cells, leading to arranged and stratified epithelial cells. Upper epithelium cells express CK13, yet the whole epithelial cells indicate ck12, representative of mucosal cell transdifferentiation [63]. In a simplified technique of SOMET, Aya Inamochi et al. transferred trypsinized pieces of oral mucosal graft on the limbal deficient surface without using any glue, AM, or suture. Soft contact lenses and tarsorrhaphy were applied to preserve the tissues at that location. Two weeks post-transplantation fluorescein staining scores and neovascularization scores were reduced. K3/K13-positive stratified epithelium was detected in the treatment group [19]. Although this method seems applicable, long-term follow-up and additional studies are needed for further confirmation. On the other hand, seeding cells requires scaffolds for settling. Tissue transplantation for recellularization without any scaffold will logically not be stable in the long term. Perhaps this novel method can serve as an alternative for scaffold-based recellularization.

## 5. Environmental and Acellularized Scaffold

Environmental signaling mimics conditions and cells to induce differentiation into target limbal cells. Although COMET seems an appropriate alternative in bilateral limbal deficiencies, post-peripheral vascularization hampers its variety of applications. When utilized as the microenvironmental factor, LNCs can limit these processes via exosomal mRNA and lead to phenotype alteration [64]. Exploitation of the 3T3 feeder cells revealed some degree of neovascularization after COMET [65]. COMECs showed lower levels of anti-angiogenetic factors and higher levels of pro-angiogenetic factors compared to cultivated corneal epithelial cells in the application of the 3T3 feeder cells [20]. The expression of the anti-angiogenic factor mRNA (PEDF and sFlt-1) increased and angiogenic mRNA factors (bFGF) decreased in the presence of the LNC compared to 3T3 in COMECs [20]. While both preserve the stemness, proliferation, and differentiation of oral mucosal epithelial cells (OMECs), insistence on LNCS competency as an ex vivo application leads to a more desirable clinical outcome [20,65]. Subconjunctival injection of the T-OMECs, compared to OMECs when co-cultured with rat LNCs, indicated less corneal epithelial defect, opacity, and neovascularization. Other markers, including cytokeratin 12 (CK12), a pigment epithelium-derived factor, and a soluble fms-like tyrosine kinase-1 also show increased levels with T-OMECs [55].

If LNCs and OMECs are co-cultured in the transwell system or the 3D Matrigel, the expression of CK12 and Pax6 may be increased [66]. All the mentioned examples demonstrated the capability of limbal cells to yield better results and induce differentiation as a co-culture factor. The limbal structure contains limbal crypts and the palisades of Vogt, which embraced the LESCs and play the pivotal role in regulating and homeostasis of the cells, which indicates how intact ECM and 3D structure fate the stem cells.

To compensate for the addressed problems and also LNC’s role in transdifferentiation and fate derivation, recent studies insist on producing the decellularized limbus to provide the best environment for cell seeding. To provide intact ECM, four techniques such as de-epithelized rabbit limbal autograft stroma, acellular porcine limbal stroma, acellular porcine corneal stroma, and de-epithelized porcine limbal stroma were evaluated by Minghai Huang et al. Two techniques including acellular porcine limbal stroma and de-epithelized rabbit limbal autograft stroma produced integrated and transparent cornea without any graft rejection. De-epithelized porcine limbal stroma was rejected in 8–10 days post-transplantation. The acellular porcine corneal stroma group maintains its K3+/P63+/Ki67+ phenotype over the 6 months [67]. Decellularized limbal graft transplantation in the limbal deficient model revealed that the limbal extracellular matrix has potential healing and re-epithelialization properties, in as soon as 7 days, and is recommended as graft transplantation in focal limbal deficiency [68]. Decellularized human limbus (DHL) was introduced by Kristina Spaniol et al. in 2017 based on sodium deoxycholate and DNase solution and γ-irradiation sterilization. Laminin, fibronectin, and collagen IV as the components of the basement membrane were preserved after decellularization and sterilization without any toxicity, presenting a well-structured ECM for LESC seeding [22]. For decellularization in the corneo-limbal region with or without limbus, three techniques such as sodium dodecyl sulfate (SDS), hypertonic saline (HS), and N2 gas (NG), were applied. However, complete decellularization in the cornea without a limbus was shown in the SDS method (0.1% SDS, 48 h), and the most severe disruption of the ECM also happened during this technique. NG showed less <50% decellularization capacity and the HS method was more favorable due to the balanced damage–decellularization ratio to the ECM [69]. Full-thickness and half-thickness porcine limbus underwent four decellularization protocols based on different combinations of SDS, double-distilled water (ddH_2_O), sodium deoxycholate, NaCl, and triton X-100. DdH_2_O for 24 h and 0.1% sodium dodecyl sulfate indicated the best preserved ECM in the half thickness of limbus, which successfully re-cellularized with human adipose-derived mesenchymal stem cells (hADSCs) and limbal epithelial cell line (SIRC), indicating the graft compatibility and potentiality in limbal regeneration [70]. The results showed that, in addition to the protocol, the thickness of the graft affects the outcome, influencing both the duration of the procedure and the severity of the protocol. A new mechanical technique assisted by microkeratomes produces thin, smooth, 360-degree limbal grafts, offering a promising approach to achieving the thinnest grafts possible while reducing the time and severity of the decellularization process [71]. Furthermore, after transplantation, this method ensures a well-matched corneo-limbal slope, facilitating centripetal cell migration [71]. These scaffolds, which mimic the limbal environment, aim to open new possibilities for limbal regeneration. However, more studies and trials are needed to further apply these methods. All the methods and results for the decellularized limbus are summarized in Table 1.

## 6. Conclusions

This review demonstrated the trend of surgical technique in bilateral limbal deficiency based on the mucosal cells and new scaffolds and how the process is becoming easier, simpler, and more affordable (Figure 1). Table 2 shows the pros and cons of different techniques using mucosal cell sources. The first method is COMET, which relies on the oral mucosal biopsy, cell suspension, and cell-culturing process. It requires extensive time, costs, and intricate steps (Figure 2A). Various steps in mucosal culturing methods have demonstrated different varieties, including scaffolds such as AM, CL, (HA + collagen), 3D tissue, fibrin, etc. As alternatives to these scaffolds, detachment factors like thermosensitive culture plates and enzymatic methods are utilized. Feeder cells for co-culturing are similarly evolving toward xeno-free materials, such as human oral mucosal fibroblasts, dermal fibroblasts, etc. The second method is mucosal graft transplantation, where the mucosal graft is directly transplanted into the deficient area (Figure 2B). Although the clinical outcomes were not desirable, this method is considered time-saving and affordable. The third approach is simple oral mucosal transplantation (SOMET), which was introduced in 2019 (Figure 2C). It was inspired by SLET in unilateral limbal deficiency. This technique led to improvements in costs, time, and ease of process. In addition to these trends, new culturing methods have demonstrated improved clinical outcomes, such as a reduction in angiogenetic factors in co-culturing LNC with mucosal cells. This concept illustrates the role of environmental signaling in differentiation. Innovative scaffolds, such as acellular limbus, create an intact ECM that can be successfully re-cellularized with limbal epithelial stem cells and hADSCs, demonstrating tissue compatibility and competency in limbal regeneration.. Perhaps the application of simple methods, such as SOMET, could serve as an innovation to utilize non-limbal cell sources and simplify the recellularization process of the scaffold.

## Figures and Tables

**Figure 1 biomedicines-13-00630-f001:**
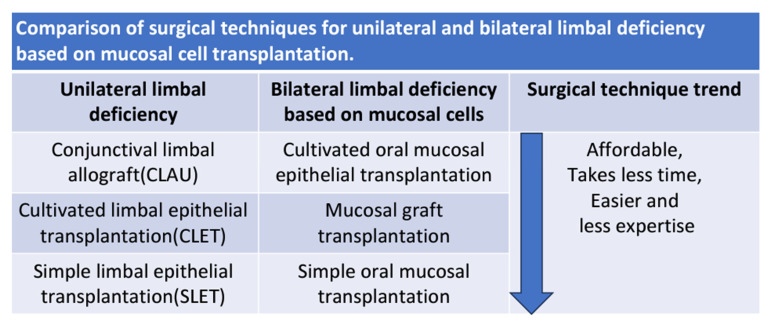
You can compare the trends of the surgical techniques in both unilateral and bilateral situations.

**Figure 2 biomedicines-13-00630-f002:**
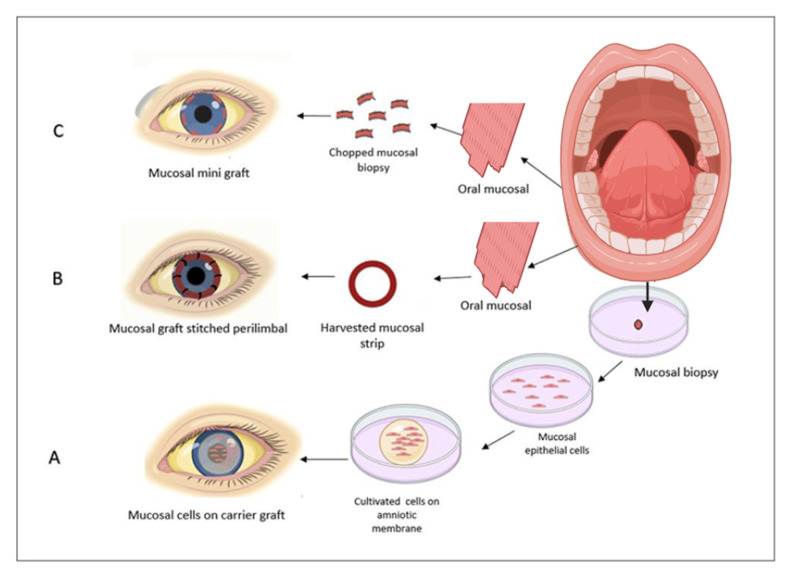
Surgical technique trends of mucosal cell transplantation. (**A**–**C**) have revealed a trend in surgical techniques based on mucosal cells. (**A**) Cultivated oral mucosal epithelial transplantation involves mechanical and enzymatic processes to create a cell suspension, as well as culturing processes to establish the cell-barrier complex necessary for transplantation. This procedure requires expertise, funding, and typically takes 2–4 weeks to complete. (**B**) Mucosal graft transplantation requires a mucosal strip that can be fixed with stitches. It requires less time, expertise, and funding, and the entire process can be completed in just a few hours. (**C**) Simple oral mucosal graft transplantation only requires a small mucosal biopsy. As you can see in the picture, these small biopsies will be fixed using fibrin glue and contact lenses, requiring minimal biopsy, less time, funding, and expertise.

**Table 1 biomedicines-13-00630-t001:** Decellularized superficial limbal graft methods, confirmation tests, the features, and outcomes associated with this procedure.

Author (Year)	Protocols	Verifications	Outcome
Minghai Huang et al. (2011) [67]	De-epithelized rabbit limbal autograft stroma (auto-DERLS) and de-epithelized porcine limbal stroma (DEPLS): 4 °C for 20 min, frozen at −70 °C for 1 h, warmed at 37 °C, and then washed by phosphate-buffered saline (PBS) with continuous shaking in a thermostat-controlled water bath for 2 h freeze–thaw (3 times). Acellular porcine limbal stroma (APLS) and acellular porcine corneal stroma (APCS): saline solution of bicarbonate mixture I (phospholipase A2 with sodium deoxycholate) for 6 h at 37 °C and followed by sodium bicarbonate mixture II (sodium deoxycholate free phospholipase A2) for 2 h at 37 °C.	1. Subcutaneous implantation 2. Transplantation on limbal defect model 3. Follow-up of transplantation models via slit lamp examination (integrity of epithelium, transparency, and neovascularization of cornea, ingrowth of conjunctival epithelial, and rejection over 6 months) 4. Histological examination (postoperative months 1, 3, and 6) 5. Immunofluorescent staining (K13, CD4 and 8, Muc5AC, goat rabbit IgG for Ki67, p63, and K3)	1. Thin fibrous around auto-DERLS, APLS, and APCS but thick fibrous around DEPLS 2. Re-epithelialization occurred in Auto-DERLS, DEPLS, and APLS in 3 to 4 days post-transplantation 3. Acute graft rejection has occurred in 8–10 days post-transplantation in DEPLS 4. The graft integration with the cornea was observed 5. APCS, APLS, and auto-DERLS groups: few T-cells (CD4+ and CD8+)
Maryam A. Shafiq et al. (2014) [68]	1. In the first step, 1.5 M NaCl solution for 48 h with a change in NaCl each 24 h. 2. In the second step, corneas were treated with both DNase and RNase (5 U/mL, Sigma-Aldrich, St. Louis, MO, USA) for 48 h. 3. Then, the corneas were washed with PBS for 72 h and PBS was changed every 24 h.	1. Slit-lamp imaging after limbal transplantation on the limbal deficient model 2. Immunohistologic evaluation (anti-p63, anti-keratin 12, and anti-Ki67)	1. A minimal amount of fluorescein staining and also stromal haziness was shown in eyes that received the graft; however, non-grafted eyes indicated peripheral neovascularization. Epithelial cells growing on the graft showed keratin 12. 2. Ki67 as a proliferation maker and DeltaNp63 as a stem cell marker were expressed more in the limbal corneal epithelial cells rather than the center of the cornea.
Kristina Spaniol (2017) [22]	1. For tissue incubation at 4 °C under continuous agitation at 200 rpm, PBS containing 5% penicillin/streptomycin was used and washed three times. 2. Decellularization solution containing 4% sodium deoxycholate monohydrate solution in a continuous agitation of 200 rpm for 48 h with changes of decellularization solution two times. 3. After washing with PBS transfer to DNase solution under continuous situation. 4. Wash with PBS under continuous agitation. 5. 25 kGy γ-irradiation used for sterilizing.	1. Using scanning electron microscopy for surface structure analysis 2. Histological characterization (H&E staining) 3. DNA content 4. Determination of the components of BM and leukocytes (fibronectin, laminin, collagen IV, CD45) 5. Cell viability assay	1. H&E (regular lamellar collagen in connective tissue). 2. 1.5 ± 0.3 μg/mg of DNA content before and 0.15 ± 0.01 μg/mg after the procedure.
Abdulkadir Isidan et al. (2021) [69]	Corneoscleral with or without limbus (n = 6), underwent 3 decellularized protocols: 1. 0.1% SDS for 48 h with continuous shaking. 2. HS (Hypertonic Saline): ultrapure water for 12 h + 30 min 2 M HS followed by 30 min ultrapure water. 3. NG (Nitrogen Gas).	1. Macroscopic assessments (transparency + severity of edema + tissue thickness) 2. H&E staining for histopathological analysis 3. Hoechst staining for DNA analysis 4. Transmission electron microscopy (TEM) for ultrastructural analysis	1. Compared to the HS and NG groups, the SDS group experienced more edema. 2. Thickness reduced in NG and SDS with limbus groups after glycerol treatment for 2 h. 3. Transparency increased in all groups after glycerol treatment for 2 h except the SDS without limbus group. 4. 94.3% and 100% of nucleic acid were successfully removed with SDS in corneas with and without limbus groups, respectively. 5. SDS disorganized lamellar collagen fiber and increased inter-fiber spacing in both with and without limbus groups.
David Sánchez-Porras et al. (2021) [70]	1. P1 Protocol: 24 h of ddH_2_O; 0.1% SDS (each 24 h: 3 incubations). 2. P2 Protocol: 24 h of ddH_2_O; 24 h of 0.1% SDS; washing via PBS; 1.5 M NaCl (each 24 h: 2 incubations). 3. P3 Protocol: 24 h of ddH_2_O; 24 h of 0.1% SDS; washing with PBS; 24 h of 1% SDC; washing with PBS; 24 h of 0.6% triton X-100; washing with PBS; 45 min of 100 mg/L DNase and 20 mg/L RNase. 4. P4 Protocol: 24 h of ddH_2_O; 24 h of 0.1% SDS; washing with PBS; 24 h of 1% SDC; washing with PBS; 24 h of 0.6% triton X-100; washing with PBS; 45 min of 100 mg/L DNase and 20 mg/L RNase; washing with PBS; 1 h of 0.05% Trypsin.	1. H&E staining 2. 4′,6-diamidino 2-phenylindole (DAPI) staining 3. Alcian blue (AB) staining (proteoglycans) 4. PSR staining 5. Historical analysis of re-cellularized limbus 6. Evaluation of p63 and pan-cytokeratin as markers of limbal cells 7. Evaluation of collagen IV and laminin (immunohistochemically) as BM components 8. PSR and AB staining for analysis of ECM components	1. <50 mg of DNA per 1 mg in all the protocols. 2. Except in protocol 1 with HL, the PSR intensity decreased; however, the area fraction increased. 3. Corneal epithelial cell numbers were high on day 7 with minimal change to days 14 and 21. Intercellular space becomes less over the 21 days. HADSC attached to DL, but on day 7, the number of cells was inadequate and increased by day 21. 4. Re-cellularized limbus with SIRC revealed p63 from day 7 to 21 and also pan cytokeratin and CRY-Z were positive at 7, 14, and 21 days. Positive p63 was shown at day 21 in hADSCs. From day 14 to 21, pan cytokeratin became positive. CRY-Z become positive from day 14 to 21. 5. At days 14 and 21, laminin was negative and collagen IV was positive in RLs with SIRC cells. RLs with hADSC were positive for laminin and collagen IV from day 14. 6. PSR area fractions were lower in two conditions of RL. In all RL than native limbus, AB-positive staining was lower.
Naresh Polisetti et al. (2021) [72]	1% SD in ultrapure water (30 min), then DPBS (3 × 30 min), then DNase I, 1 mg/mL (overnight), and then washed for 4 × 30 min in DPBS. All of the steps were carried out under 4% dextran.	1. H&E staining 2. Human leukocyte antigen (HLA-ABC) 3. DAPI 4. Light and electron microscopy for evaluating limbal architecture 5. Periodic acid Schiff (PAS) and AB staining to evaluate ECM Components (collagen III, IV, and XVIII, again, junctional adhesion molecule C (JAM-C), tenascin C (TN-C), fibronectin (FN), laminin (LN) chains (α3, α5, β2, β3, and γ2), and vitronectin (VN)) 6. In vitro recellularization (limbal epithelial progenitor cells (LEPC) and limbal melanocytes (LM)): intercellular E (epithelial)-cadherin, pan-cytokeratin, vimentin, p63, CK15, Ki-67, CK3, and Melan-A staining 7. Ex vivo transplantation (pan-cytokeratin, CK3, p63, Ki-67, Melan-A, and vimentin staining)	1. Multilayer epithelium, dark stained cells in the basal layer, arranged, and regular collagen fiber. 2. No cellular and material debris was found in DL. 3. Removed about 98.5 ± 0.3% with dextran and 99.2 ± 0.4% w/o it. 4. Stromal projections demonstrated in DL in both w or w/o. Epithelial basement membrane significantly detected in dextran-treated DHL. 5. Re-cellularized scaffolds phenotype: E-cadherin and pan-cytokeratin in all of the epithelial layers, p63 and CK15 in the basal layer, Ki-67 in the basal limbal layer, vimentin in the basal and stromal layer, and Melan-A in the epithelial layer. 7. Pan-cytokeratin positive in host and graft cells and CK3 positive cells in superficial and all of the epithelial layers in the graft and host, respectively. CK15 presents on the basal epithelial cells of the graft, however, was not positive in the host corneal cells. The basal layer of the graft and also a few cells on the host were positive for P63. Ki-67 was positive in the basal layer and negative in the host tissue. Melan-A positive melanocytes were presented at the corneal region.

**Table 2 biomedicines-13-00630-t002:** Comparison of the pros and cons of surgical techniques based on mucosal cells.

Simple Oral Mucosal Transplantation (SOMT)	Oral Mucosal Graft Transplantation (OMGT)	Cultured Oral Mucosal Cell Transplantation (COMCT)
Pros	Cons and Future Considerations	Pros	Cons and Future Considerations	Pros	Cons and Future Considerations
Needs fewer facilities	Requires more and long-term clinical trials	Needs fewer facilities	Needs more tissue grafts to create the annular or crescent graft	Many in vitro, in vivo, and clinical trials have been conducted	Requires expertise and facilities
All the procedures are performed beside the patients in OR	More innovation in Auto-SLET should be assessed, considering the suitable base membrane, adding AM as scaffolds, or enhancing healing	All procedures are performed beside the patients in the operation room	Requires more and long-term clinical trials		Take 2 to 4 weeks for culturing process
Time saving	The consideration of other cell source, such as Auto-SLET should be compared	Time saving			Expensive and not cost-effective
Cost benefits		Cost benefits			
Reduced the need for tissue grafts for transplantation					
Reduced the induction of limbal deficiency

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
