# Peer review of "Advancing Bilateral Limbal Deficiency Surgery: A Comprehensive Review of Innovations with Mucosal Cells"

_biomedicines, 2025, doi:10.3390/biomedicines13030630_

Round 1

Reviewer 1 Report

Comments and Suggestions for Authors

The manuscript titled “Advancing Bilateral Limbal Deficiency Surgery: Innovations with Mucosal Cells and Acellularized Limbus Scaffolds; A Comprehensive Review” demonstrated the trend of surgical technique in bilateral limbal deficiency based on the mucosal cells. However, lack of coherence is the major flaw of the study. Therefore, minor revision has to be done before this manuscript could be accepted for publication in the Biomedicines.

Major comments:

In the introduction section, the authors need to provide detailed information on current progress in surgeries, the article describes a lot about cultivating oral mucosal epithelial sheets.

Minor comments:

1. Line 95, the article does not seem to address “explore” new scaffold materials.

2. Line 97-99, please add reference(s).

3. Line 127-131, This seems unrelated to “Cultivated oral mucosal epithelial sheets”.

4. Line 230, LNCs are mentioned for the first time in the text, so the abbreviation should not be used.

5.  The conclusion does not mention the prospective and limitations.

Author Response

Major comments:

In the introduction section, the authors need to provide detailed information on current progress in surgeries, the article describes a lot about cultivating oral mucosal epithelial sheets.

Thank you for your thoughtful comments. We have interpolated more details about the mucosal culture sheet to emphasize that the technique is long-term, expensive, and time-consuming. Additionally, I have already added more details about other techniques. This review is entirely based on the mucosal cell source technique, with most of the innovations interpreted and discussed in the text. Green highlight

Minor comments:

  1. Line 95, the article does not seem to address “explore” new scaffold materials.

We have rewritten this section. Line 111-113. Purple highlight

  1. Line 97-99, please add reference(s). Done, line 117

  1. Line 127-131, This seems unrelated to “Cultivated oral mucosal epithelial sheets”.

Thank you for your comments. These references emphasize the clinical outcomes of these methods after transplantation. I didn’t understand what you meant by 'unrelated.

  1. Line 230, LNCs are mentioned for the first time in the text, so the abbreviation should not be used. Done

  1.   The conclusion does not mention the prospective and limitations. Blue lines 408-413

Reviewer 2 Report

Comments and Suggestions for Authors

This manuscript was review of comparison techniques in treatment of stem cells deficiency. The introduction was ok and acceptable , in addition the other parts were well-written and comprehensive indeed. I want to ask the authors to add one table and summarise all techniques in it and also mentioned advantages and disadvantages of each techniques in another table.I recomend the authors to add a picture for each techniques. 

Author Response

Thank you for your comments. We have attached a comparison table of different surgical techniques. Figures 1 and 2, which we uploaded earlier, are representative of the different surgical procedures based on mucosal cells.

Reviewer 3 Report

Comments and Suggestions for Authors

I think this review has well-organized insights to summarize the current techniques for bilateral limbal deficiency.

This is a review related with current techniques involving mucosal cells include cultivated oral mucosal transplantation (COMT), oral mucosal graft transplantation (OMGT), and simple oral mucosal transplantation (SOMT). COMT requires suspension of cells and a culturing process that is time-consuming and cost-prohibitive. Although some undesirable outcomes, such as angiogenesis, can occur post-trans-plantation, and the ultimate goal of differentiation into limbal epithelial stem cells may not be achieved, mucosal cell sources can be a good alternative for stabilizing the ocular surface. This review demonstrates the ongoing changes in surgical technique trends and how they have made mucosal cell transplantation easier and more effective and acellularized limbus as a scaffold for limbal regeneration. I think this review has well-organized insights to summarize the current techniques.

Author Response

Thank you for your thoughtful comments.